# Mechanisms involved in suppression of osteoclast supportive activity by transforming growth factor-β1 via the ubiquitin-proteasome system

**Momoko Inoue[1,2], Yoshie Nagai-Yoshioka[1], Ryota Yamasaki[1], Tatsuo Kawamoto[2], Tatsuji Nishihara[1], Wataru Ariyoshi[1] ***

**1** Division of Infections and Molecular Biology, Department of Health Promotion, Kyushu Dental University, Kitakyushu, Fukuoka, Japan, **2** Division of Orofacial Functions and Orthodontics, Department of Health Promotion, Kyushu Dental University, Kitakyushu, Fukuoka, Japan

* arikichi@kyu-dent.ac.jp

**Data Availability Statement:** All relevant data are within the paper and its Supporting information files.

## Abstract

Orthodontic treatment requires the regulation of bone remodeling in both compression and tension sides. Transforming growth factor-β1 (TGF-β1) is an important coupling factor for bone remodeling. However, the mechanism underlying the TGF-β1-mediated regulation of the osteoclast-supporting activity of osteoblasts and stromal cells remain unclear. The current study investigated the effect of TGF-β1 on receptor activator of nuclear factor kappa-B ligand (RANKL) expression in stromal cells induced by $1\alpha,25(OH)_2D_3$ ($D_3$) and dexamethasone (Dex). TGF-β1 downregulated the expression of RANKL induced by $D_3$ and Dex in mouse bone marrow stromal lineage, ST2 cells. Co-culture system revealed that TGF-β1 suppressed osteoclast differentiation from bone marrow cell induced by $D_3$ and Dex-activated ST2 cells. The inhibitory effect of TGF-β1 on RANKL expression was recovered by inhibiting the interaction between TGF-β1 and the TGF-β type I/activin receptor or by downregulating of smad2/3 expression. Interestingly, TGF-β1 degraded the retinoid X receptor (RXR)-α protein which forms a complex with vitamin D receptor (VDR) and regulates transcriptional activity of RANKL without affecting nuclear translocation of VDR and phosphorylation of signal transducer and activator of transcription3 (STAT3). The degradation of RXR-α protein by TGF-β1 was recovered by a ubiquitin-proteasome inhibitor. We also observed that poly-ubiquitination of RXR-α protein was induced by TGF-β1 treatment. These results indicated that TGF-β1 downregulates RANKL expression and the osteoclast-supporting activity of osteoblasts/stromal cells induced by $D_3$ and Dex through the degradation of the RXR-α protein mediated by ubiquitin-proteasome system.

## Introduction

Bone is constantly renewed and maintained via the balance between osteoclastic bone resorption and osteoblastic bone formation. This repetitive process, termed "bone remodeling",

**Funding:** This study was partially supported by the Japan Society for the Promotion of Science Grant-in-Aid for Scientific Research awarded to WA (grant number: 18K09797 and 21H03145). No additional external funding was received for this study.

**Competing interests:** The authors have declared that no competing interests exist.

plays important roles in the development and maintenance of skeletal tissue [1, 2]. Mechanical stress is an important factor in bone remodeling because component cells of bone are sensitive and responsive to mechanical loading [3]. In the orthodontic tooth movement, osteoclasts cause resorption of alveolar bone on the side which is subjected to compression, while osteoblasts induce bone formation on the side subjected to tension [4, 5]. Regulation of bone remodeling on both compression and tension sides plays a major role in determining the success of the orthodontic procedure.

Osteoclasts are multinucleated giant cells derived from precursor cells stemming from a monocyte/macrophage lineage [6]. Receptor activator of nuclear factor κ-B ligand (RANKL; encoded by the *Tnfsf11* gene) is essential for osteoclastogenesis and regulates bone remodeling and calcium homeostasis. In bone tissue, RANKL is expressed on the surface of stromal cells, osteoblasts, and their precursors [7]. RANKL directly bind to a type I transmembrane protein (RANK), highly expressed in the membrane of osteoclast progenitors and mature osteoclasts [8]. RANKL-RANK system activates several signaling pathways, indispensable for osteoclast formation. Osteoprotegerin (OPG), a member of the tumor necrosis factor (TNF) receptor superfamily, is known as a decoy receptor of RANKL and protects bone from excessive resorption by binding to RANKL and preventing it from binding to RANK [2].

As an active metabolite of vitamin $D_3$, $1\alpha,25(OH)_2D_3$ ($D_3$) stimulates RANKL expression in osteoblasts and stromal cells via vitamin D receptor (VDR) [9, 10]. $D_3$ activates a complex formation between VDR and retinoid X receptor (RXR)-$\alpha$, a member of the steroid receptor family. The heterodimerized VDR-RXR complex shows strong affinity for binding with vitamin D response elements (VDREs) and regulates transcriptional activity [11].

Transforming growth factor-β (TGF-β) family members consisting of TGF-β, bone morphogenetic proteins (BMP), and activin have been shown to be important coupling factors for bone formation and bone resorption. Previous studies reported that TGF-β1 was highly expressed on the periodontal tissue such as the periodontal ligament cells and the surface of the alveolar bone during orthodontic tooth movement, suggesting that TGF-β1 may contribute to bone remodeling process [12–15]. However, the mechanisms underlying the regulation of osteoclast-supporting activity of osteoblasts and stromal cells by TGF-β1 are unknown. The objectives of this study were to elucidate the effects of TGF-β1 on the RANKL expression induced by $D_3$ in mouse-derived stromal cell.

## Materials and methods

### Reagents and antibodies

Recombinant human TGF-β1 and the mouse TRANCE/TNFSF11/RANKL polyclonal antibody were purchased from R&D Systems (Minneapolis, MN, USA, #AF462). Dexamethasone (Dex), $D_3$, and anti-β-actin monoclonal antibodies (#1978) were purchased from Sigma Aldrich (St. Louis, MO, USA). Mouse VDR monoclonal antibody was obtained from Santa Cruz Biotechnology (Santa Cruz, CA, USA, #sc-13133), and anti-signal transducer and activator of transcription3 (STAT3) monoclonal (#4904), anti-phospho-STAT3 polyclonal (#9131), anti-RXR-$\alpha$ monoclonal (#3085), anti-ubiquitin monoclonal (#3936), and anti-Histone H3 polyclonal antibodies (#9715) were purchased from Cell Signaling Technology (Beverly, MA, USA).

### Cell culture

Mouse-derived stromal cell line, ST2 (Riken Cell Bank, Ibaraki), was cultured and maintained in α-minimum essential medium (α-MEM) (Gibco, Grand Island, NY, USA) supplemented with 10% fetal bovine serum (FBS) (Sigma-Aldrich), 100 units/mL penicillin G potassium salt,

and 100 μg/mL streptomycin at 37˚C in an atmosphere of a 5% $CO_2$. This cell line shows high expression of RANKL and low expression of OPG when treated with $D_3$ and Dex. Cells ($5 \times 10^5$ cells/well) seeded for overnight in a 6-well plate were treated with $D_3$ ($10^{-7}$ M), Dex ($10^{-7}$ M), and TGF-β1. In some experiments, activation of smad2/3 was inhibited by pre-treatment with ALK4/5/7 inhibitor (A83-01; Tocris Bioscience, Bristol, UK) for 1 h. MG132 (Calbiochem-Novabiochem Corp, San Diego, CA, USA) was used as an inhibitor for protein degradation by ubiquitin-proteasome system.

## Quantitative real-time reverse transcriptase (RT)-PCR analysis

Total RNA was isolated from ST2 cells using a Cica Geneus RNA Prep Kit (Kanto Chemical, Tokyo, Japan) according to the manufacturer instructions. Reverse transcribed into cDNA and quantitative PCR reaction were performed following a previous reported protocol [16]. Total cDNA abundance in samples was normalized to *gapdh* mRNA expression. The mRNA expression of *rankl*, *opg*, *cyp24a*, *smad2/3*, and *rxr-α* were quantitated via real-time RT-PCR using the following primer sequences: *gapdh*, 5′–GACGGCCGCATCTTCTTGA–3′ (forward) and 5′–CACACCGACCTTCACCATTTT–3′ (reverse); *rankl*, 5′–GGCCACAGCGCTTCTCA–3′ (forward) and 5′–CCTCGCTGGGCCACATC–3′ (reverse); *opg*, 5′–GCCTGGGACCAAAGTGAATG–3′ (forward) and 5′–GACATTCGAGGCTCCAGTGAA–3′ (reverse); *cyp24a1*, 5′–ACAGCGAGCTGAACAAATGG–3′ (forward) and 5′–TTTGATGGCCGCAATGAAGG–3′ (reverse); *smad2*, 5′–ACACAACAGGCCTTTACAGC–3′ (forward) and 5′–AACATGTGGCAACCCTTTCCC–3′ (reverse); *smad3*, 5′–AGAACGTGAACACCAAGTGC–3′ (forward) and 5′–AATTCATGGTGGCGCAGTGT–3′ (reverse); *rxr-α*, 5′–AGATGCGTGACATGCAGATG–3′ (forward) and 5′–TGCAGTACGCTTCTAGTGACG–3′ (reverse).

## Western blot analysis

Whole cell lysates were extracted using a Cell Lysis Buffer (Cell Signaling Technology) containing protease inhibitor (Thermo Scientific, Rockford, IL, USA). Western blot analysis was carried out as previously described [16] with primary antibodies (1:1000) against RANKL, VDR, STAT3, phospho-STAT3, RXR-α, and β-actin. In some experiments, nuclear and cytoplasmic fractions were isolated using a NE-PER kit from Thermo Scientific according to the manufacturer's instructions. Gel images from independent triplicate analyses were subjected to densitometric analysis using Image LabTM® 2.0 software (Bio-Rad, Hercules, CA, USA). The relative band intensity values were normalized to changes in the β-actin, total protein or Histone H3 of the same sample.

## Mouse bone marrow cell isolation

Mouse bone marrow cells (BMCs) were extracted from tibias and femurs of 6-week-old ddY male mice (Japan SLC, Inc. Shizuoka, Japan). BMCs were seeded in 10-cm plates and cultured for 3 days in α-MEM with 10% FBS and 20 ng/ml recombinant human macrophage colony-stimulating factor (rhM-CSF) (Rocky Hill, NJ, USA). Adherent cells were used as bone marrow-derived macrophages (BMMs). All the procedures were approved by the Animal Care and Use Committee of Kyushu Dental University (Protocol Number: 20–23). All surgery was performed under sodium pentobarbital anesthesia, and all efforts were made to minimize suffering.

## Osteoclast formation assay using co-culture system

ST2 cells ($6 \times 10^3$ cells/well) were plated in 48-well plates and maintained in α-MEM with 10% FBS, $D_3$ ($10^{-7}$ M), Dex ($10^{-7}$ M), and TGF-β1 (2.0 ng/ml) for 24 h. Stimulated ST2 cells were

co-cultivated with BMMs ($1 \times 10^5$ cells/well). This co-cultivation system was maintained in α-MEM with 10% FBS in the presence of D$_3$ ($10^{-7}$ M), Dex ($10^{-7}$ M), and TGF-β1 (2.0 ng/ml), with a medium change every 2 days. On 4 days, to detect osteoclast formation, TRAP staining was performed by using a leukocyte acid phosphatase kit (Sigma-Aldrich), according to the manufacturer's protocol. TRAP-positive multinucleated cells containing more than three nuclei were defined as osteoclasts.

## Silencing of smad2/3 expression by small interfering RNA (siRNA) transfection

Specific siRNA (Santa Cruz Biotechnology) was used to knockdown smad2 and smad3 expression in ST2 cells. Control siRNA was also obtained from Santa Cruz Biotechnology. Super Electroporator NEPA21 and electroporation cuvettes (Nepa Gene, Chiba, Japan) were used to transfect 1.65 μL of siRNA into ST2 cells ($1.0 \times 10^6$ cells/mL) plated in 6-well plates. Transfected ST2 cells were incubated for 24 h prior to stimulation with D$_3$ ($10^{-7}$ M), Dex ($10^{-7}$ M), and TGF-β1 (2 ng/mL).

## Co-immunoprecipitation

ST2 cells were pre-treated with MG132 (5 μM) for 1 h prior to treatment with TGF-β1 (2 ng/mL). Whole cell lysates were prepared using Cell Lysis Buffer containing protease inhibitor. For immunoprecipitations, 1.5 mg of magnetic Dynabeads® Protein G (Invitrogen) were conjugated with anti-RXR-α antibodies (5 μg), followed by the incubation with the whole cell lysate for 60 min at room temperature according to the previous reported protocol [17]. The antigens were eluted from Dynabeads®-antibody-antigen complex by 50 mM glycine, (pH 2.8) and sodium dodecyl sulfate (SDS) sample buffer with β-mercaptoethanol. After the heating for 10 min at 70˚C, eluted samples were subjected to SDS polyacrylamide gel electrophoresis and Western blot analysis using the anti-ubiquitin monoclonal antibody.

## Immunofluorescence microscopy

ST2 cells were seeded at $1 \times 10^4$ cells per well in 4-well chamber slides (Thermo Scientific). After a 24 h attachment period, cells were treated with D$_3$ ($10^{-7}$ M), Dex ($10^{-7}$ M), and TGF-β1 (2 ng/mL) for indicated time periods. Cells were fixed with 4% paraformaldehyde for 20 min at room temperature and washed by 0.2 M glycine in phosphate-buffered saline (PBS). 10-min treatment with 0.2% Triton X-100 was used to permeabilize fixed cells. Nonspecific binding sites were blocked for 30 min by 1% bovine serum albumin (BSA) (Sigma-Aldrich) in PBS. The cells were then incubated with anti-TRANCE/TNFSF11/RANKL polyclonal, anti-RXR-α monoclonal, and anti-ubiquitin monoclonal antibodies overnight at 4˚C. After washing with PBS, the cells were incubated with Alexa Fluor® 488-conjugated donkey anti-goat IgG, Alexa Fluor® 488-conjugated goat anti-rabbit IgG, and Alexa Fluor® 568-conjugated donkey anti-mouse IgG (Invitrogen) for 1 h at room temperature, and washed and mounted in nuclear staining agent 4′, 6-diamino-2-phenylindole (DAPI). Image processing and analysis were performed using fluorescence microscope (BZ-9000, Keyence, Osaka, Japan).

## Statistical analysis

All experiments were replicated thrice and data are expressed as the mean ± standard deviation of three independent experiments. Statistical analyses were conducted using JMP® software, version 10.0.2 (SAS Institute Inc., Cary, NC, USA) and analyzed via one-way analysis of

variance (ANOVA) followed by a suitable post hoc test (Dunnett's or Tukey's) for multiple comparison. Statistically significant was set at $P < 0.05$.

## Results

### TGF-β1 suppresses RANKL expression induced by $D_3$ and Dex

To elucidate the effects of TGF-β1 on the RANKL and OPG expression induced by $D_3$ and Dex in stromal cells, ST2 cells were cultured with $D_3$, Dex, and TGF-β1. The mRNA expression of *rankl* induced by $D_3$ and Dex was significantly suppressed by the addition of TGF-β1 (Fig 1A). However, the down-regulation of *opg* mRNA by $D_3$ and Dex was not aeffected by the addition of TGF-β1 (Fig 1B). Real-Time RT-PCR revealed that the expression of *cyp24a1*, $1\alpha,25(OH)_2D_3$ responsive gene, stimulated by $D_3$ and Dex was strongly downregulated by TGF-β1 (Fig 1C). This inhibitory effect of RANKL expression by TGF-β1 was observed in protein level (Fig 1D). Moreover, immunofluorescence staining revealed that RANKL protein accumulation on the cell surface induced by $D_3$ and Dex was diminished by TGF-β1 treatment (Fig 1E).

### TGF-β1 suppresses osteoclast differentiation of BMMs cells co-cultured with ST2 cells stimulated by $D_3$ and Dex

To clarify whether TGF-β1 possesses biological effect on osteoclast formation supported by ST2 cells, ST2 cells were treated with $D_3$, Dex and TGF-β1, and then co-cultured with BMMs. After 4 days cultivation, the formation of osteoclasts was assessed using TRAP staining. We found that treatment of ST2 cells with TGF-β1 suppressed the differentiation of BMMs into multinucleated osteoclasts induced by $D_3$ and Dex (Fig 2).

### Pre-treatment with A83-01 and knockdown of smad2/3 recovers the inhibitory effect of TGF-β1 on RANKL expression

To examine the interaction between TGF-β1 and the TGF-β type I/activin receptor on RANKL expression induced by $D_3$ and Dex, ST2 cells were pretreated with A83-01 for 1 h. A83-01 recovered the suppression of RANKL mRNA (Fig 3A) and protein (Fig 3B) expression by TGF-β1. Next, to further determine the role of TGF-β1 in RANKL expression induced by $D_3$ and Dex, control siRNA and smad2/3 siRNA were introduced into ST2 cells in order to knockdown smad2/3 expression. The introduction of specific siRNA effectively inhibited *smad2* and *smad3* mRNA expression (Fig 3C). The inhibitory effect of TGF-β1 on *rankl* mRNA expression induced by $D_3$ and Dex tended to be recovered in ST2 cells transfected with smad2 and smad3-specific siRNA (Fig 3D). Especially, mRNA of *smad2* was in larger part responsible for the recovery of *rankl* expression.

### TGF-β1 has no effect on activation of signaling pathway mediated by nuclear VDR or STAT3

RANKL expression is regulated by several transcription factors such as VDR and STAT3 [18]. First, we elucidated the effect of TGF-β1 on activation of VDR by Western blotting. Pre-treatment of TGF-β1 had no effect on the nuclear translocation of VDR induced by $D_3$ and Dex (Fig 4A). Moreover, pre-treatment of TGF-β1 did not affect the phosphorylation level of STAT3 protein (Fig 4B).

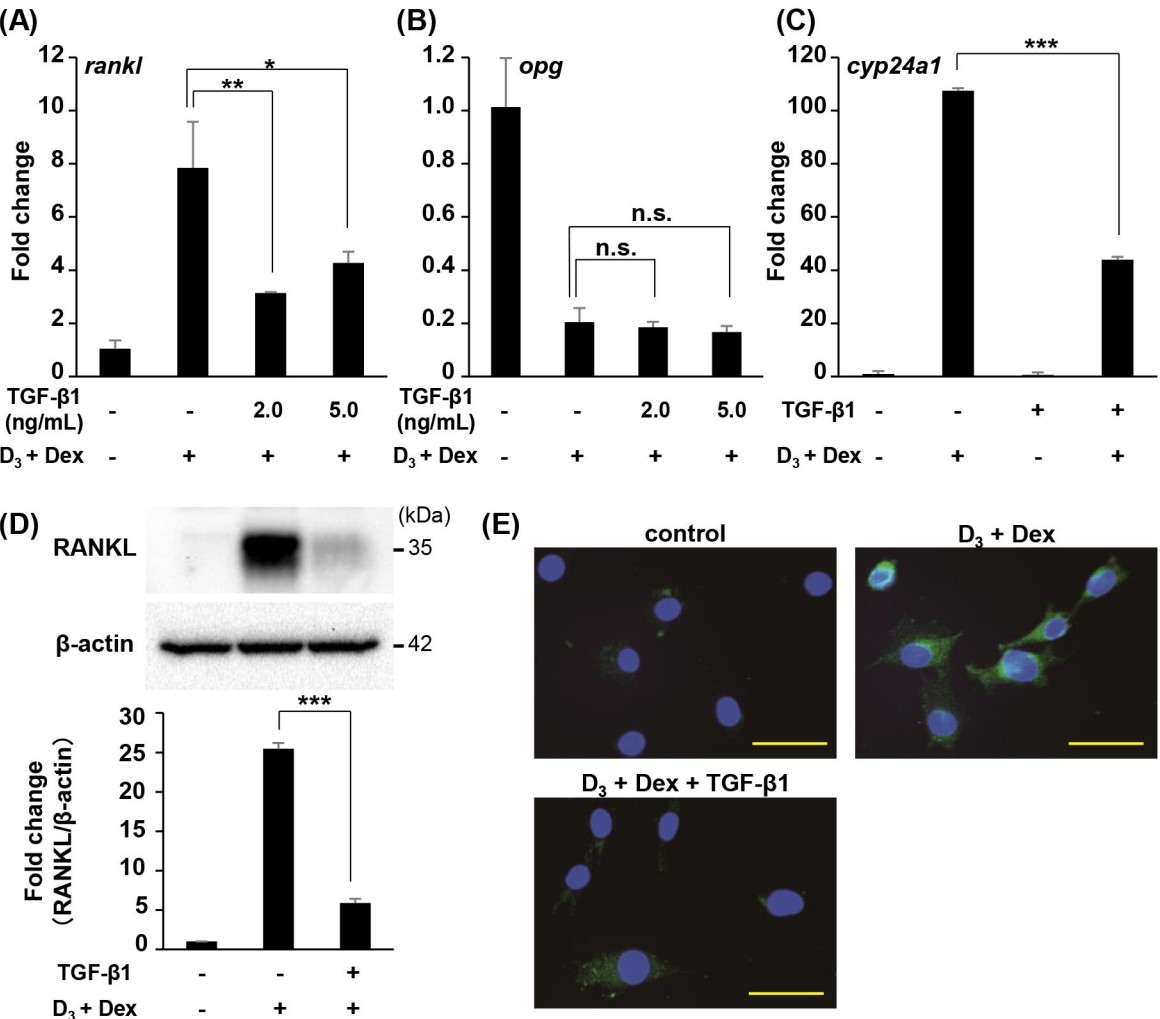

**Fig 1. Effect of TGF-β1 on RANKL expression induced by 1α,25(OH)₂D₃ and dexamethasone in ST2 cells.** (A, B) ST2 cells were treated with 1α,25(OH)₂D₃ (D₃) ($10^{-7}$ M), dexamethasone (Dex) ($10^{-7}$ M), and TGF-β1 (2.0, 5.0 ng/mL) for 12 h. (C) ST2 cells were treated with D₃ ($10^{-7}$ M), Dex ($10^{-7}$ M), and TGF-β1 (2.0 ng/mL) for 12 h. The mRNA level of *rankl* (A), *opg* (B), and *cyp24a1* (C) were measured using real-time RT-PCR. (D, E) ST2 cells were treated with D₃ ($10^{-7}$ M), Dex ($10^{-7}$ M), and TGF-β1 (2.0 ng/mL) for 24 h. (D) The protein level of RANKL was detected using Western blot analysis. β-actin served as the loading control. Bars represent means with the standard deviation of relative band intensities normalized to changes in the β-actin from independent triplicate samples. (E) Cells were fixed and immunostained using a polyclonal antibody to detect RANKL (green). Blue represents DAPI staining. Scale bars indicate 50 μm. Graph data are displayed as mean ± S.D. of three independent cultures. *n.s.* = not significant, * = $P < 0.05$, ** = $P < 0.01$, *** = $P < 0.0001$.

## TGF-β1 downregulates the RXR-α protein expression without affecting mRNA expression

To further examine the molecular mechanisms by which TGF-β1 suppresses RANKL expression, we focused on RXR-α which heterodimerized with VDR. The expression of RXR-α protein was strongly inhibited by TGF-β1 in time dependent manner up to 12 h (Fig 5A). Interestingly, TGF-β1 had almost no effect on the mRNA expression of *rxr-α* (Fig 5B). Suppression of RXR-α protein by TGF-β1 was recovered by A83-01 pre-treatment (Fig 5C) and smad2/3 siRNA introduction (Fig 5D).

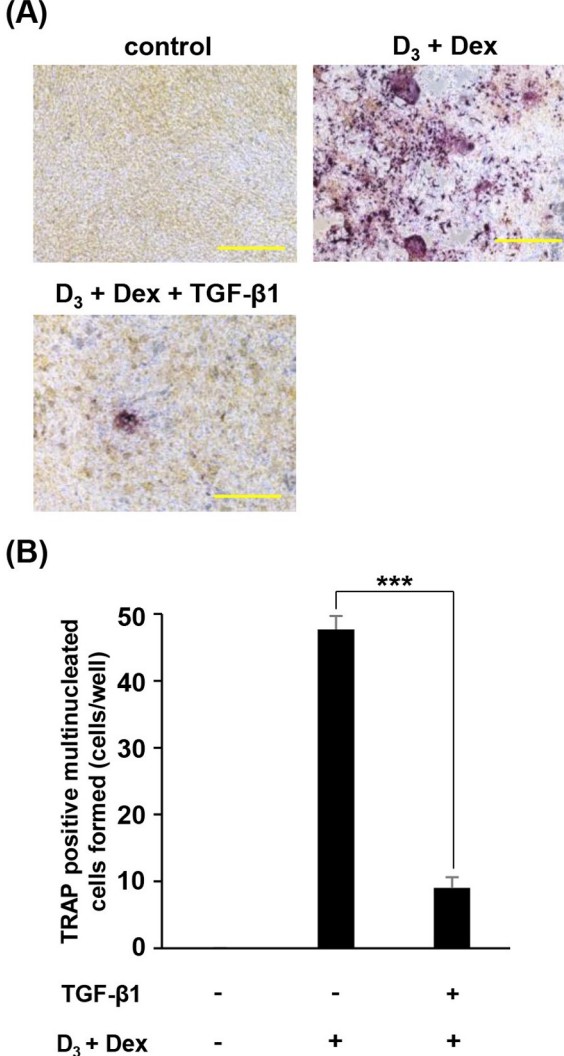

**Fig 2. Effect of TGF-β1 on osteoclast formation induced by 1α,25(OH)$_2$D$_3$ and dexamethasone-stimulated ST2 cells.** ST2 cells were treated with 1α,25(OH)$_2$D$_3$ (D$_3$) ($10^{-7}$ M), dexamethasone (Dex) ($10^{-7}$ M), and TGF-β1 (2.0 ng/mL) for 24 h. After 24 h, ST2 cells were co-cultured with BMMs for 4 days. (A) TRAP analysis was performed to detect osteoclast formation. Scale bars indicate 300 μm. (B) TRAP-positive cells containing more than three nuclei were counted. Data are expressed as the mean ± S.D. of triplicate cultures. ***$P < 0.0001$.

## TGF-β1 induces RXR-α protein degradation mediated by ubiquitin-proteasome system

To fully investigate the proteolytic system of RXR-α induced by TGF-β1, ST2 cells were pretreated with proteasome inhibitor (MG132). Suppression of RXR-α protein by TGF-β1 was recovered by MG132 (Fig 6A). Then, we examined the ubiquitination of RXR-α by co-immunoprecipitation and found that the level of ubiquitinated RXR-α protein by TGF-β1 was reached the peak at 3 h (Fig 6B). Immunofluorescence microscopic analysis also revealed that RXR-α accumulated largely in the nuclei of ST2 cells and its colocalization with ubiquitin was increased by TGF-β1 stimulation for 3 h (Fig 6C).

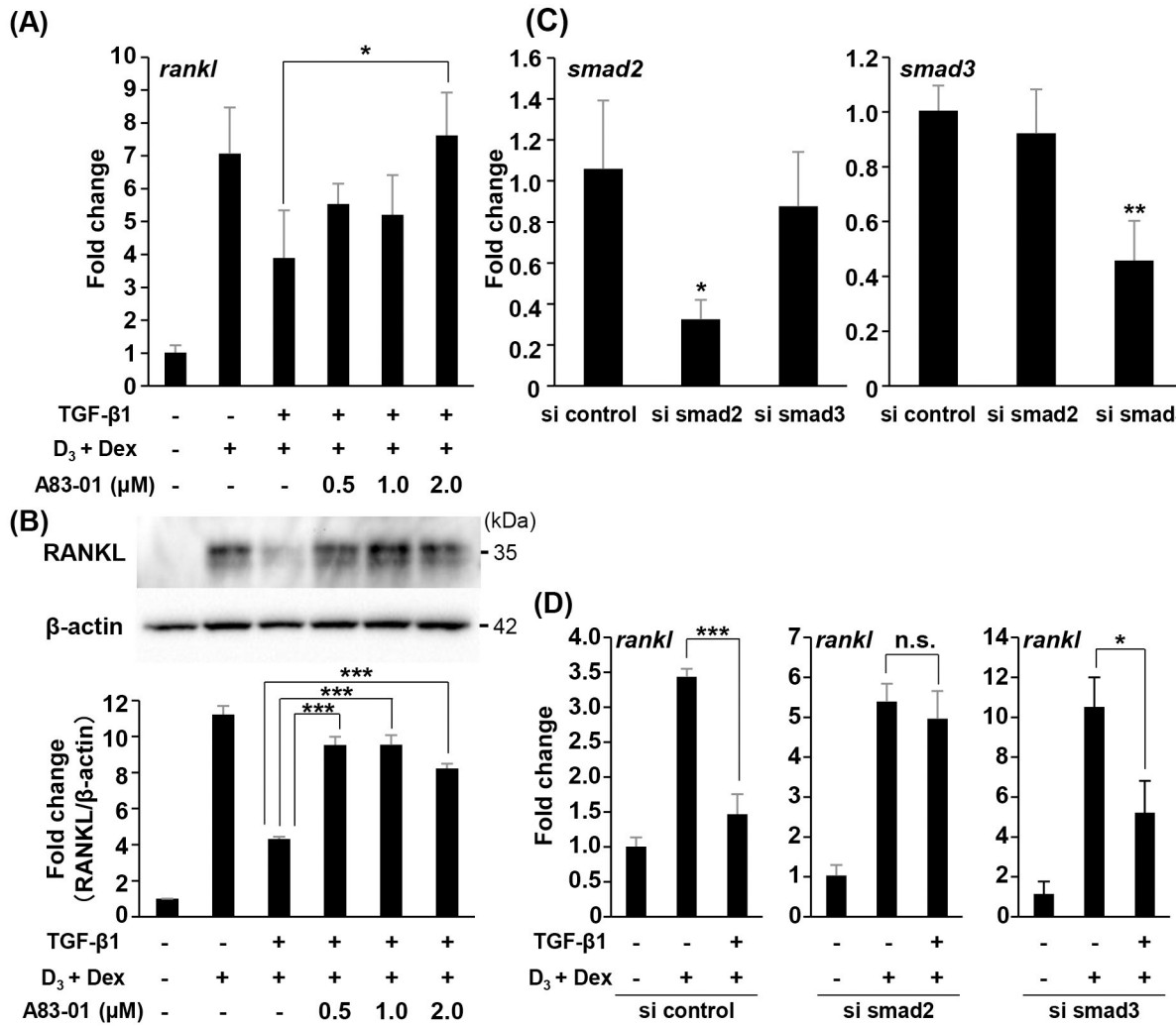

**Fig 3. Effect of A83-01 and specific siRNA knockdown of smad2/3 on RANKL expression in ST2 cells.** (A, B) ST2 cells were pretreated with indicated concentrations of A83-01 for 1 h. Cells were then incubated with $1\alpha,25(OH)_2D_3$ (D$_3$) ($10^{-7}$ M), dexamethasone (Dex) ($10^{-7}$ M), and TGF-β1 (2.0 ng/mL) for 12 h (A) and 24 h (B). The mRNA (A) and the protein (B) level of RANKL were detected using real-time RT-PCR and Western blot analysis, respectively. (B) Bars represent means with the standard deviation of relative band intensities normalized to changes in the β-actin from independent triplicate samples. (C) ST2 cells were transfected with either control siRNA or smad2/3 siRNA and then cultured for 24 h. The mRNA levels of *smad2* and *smad3* were assessed using real-time RT-PCR. (D) Transfected ST2 cells were incubated for 24 h, and stimulated with D$_3$ ($10^{-7}$ M), Dex ($10^{-7}$ M), and TGF-β1 (2.0 ng/mL) for 12 h. The mRNA level of *rankl* was assessed using real-time RT-PCR. Data are displayed as mean ± S.D. of three independent cultures. *n.s.* = not significant, $^*P < 0.05$, $^{**}P < 0.01$, $^{***}P < 0.0001$.

## Discussion

TGF-β1 is known to involve in the differentiation of a variety of cells such as osteoblasts and plays an important role in regulating physiological processes including bone remodeling. Although several studies demonstrated that the role of TGF-β1 on bone remodeling process, the reported effects of TGF-β1 are controversial. While some studies have showed the enhancement of osteoclast differentiation by TGF-β1 [19–21], other researchers have reported opposing effects [22–24]. Since many of these studies are focused on the direct effects of TGF-β1 on osteoclast progenitors, we investigated the effects of TGF-β1 on stromal cells which generate RANKL and induce osteoclast differentiation. To elucidate the effect of TGF-β1 on

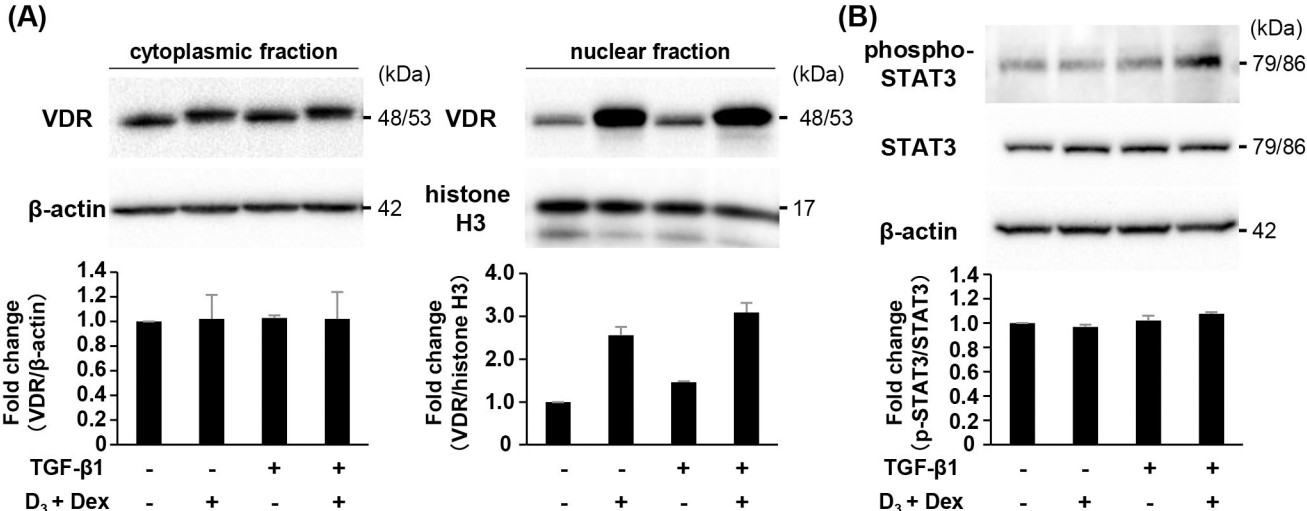

**Fig 4. Effect of TGF-β1, D₃ and Dex on nuclear translocation of VDR and phosphorylation of STAT3.** ST2 cells were treated with 1α,25(OH)₂D₃ (D₃) ($10^{-7}$ M), dexamethasone (Dex) ($10^{-7}$ M), and TGF-β1 (2.0 ng/mL) for 2 h (A) and 1 h (B). (A) Nuclear and cytoplasmic fractions were isolated and subjected to Western blot analysis for VDR. β-actin (cytoplasmic fraction) and histone H3 (nuclear fraction) served as the loading control. (B) Whole cell lysates were subjected to Western blot analysis for STAT3 and phospho-STAT3. β-actin served as the loading control. Bars represent means with the standard deviation of relative band intensities normalized to changes in the β-actin, histone H3, and STAT3 from independent triplicate samples.

osteoclast-supportive activity, we used mouse bone marrow stromal lineage ST2 cells, which are known to express RANKL via D₃ and support the osteoclast differentiation [25]. This study design did not contain any osteoclast precursors, which may also become potential targets of TGF-β1 [26].

Expression of RANKL induced by D₃ and Dex was obviously decreased by the addition of TGF-β1. Moreover, the osteoclast formation from BMMs by D₃ and Dex -activated ST2 cells was decreased by addition of TGF-β1. On the other hand, TGF-β1 had no effect on the expression of OPG which prevent RANKL/RANK interaction and osteoclast differentiation. These observations suggest that TGF-β1 exerted modulative effects against osteoclastogenesis by inhibiting RANKL expression via downregulation of cellular signaling activated by D₃. These data are consistent with the previous data which demonstrated that high concentration of TGF-β suppresses the expression of RANKL in osteoblasts [26, 27]. The gene expression of *cyp24a1*, one of the hydroxylase enzymes which regulates D₃ catabolism [28, 29] induced by D₃ and Dex was also suppressed by TGF-β1.

TGF-β1 binds to serine-threonine kinase receptors type I (TβRI, ALK5) and type II receptor (TβRII), which form a hetero-tetramer that specifically induces phosphorylation of the receptor activate (R-) smads such as smad2 and smad3 [30]. RANKL expression, which had been inhibited by TGF-β1, was significantly recovered by pre-treatment with inhibitor of TβRI and knockdown of smad2 and smad3 gene. These findings indicated that negative regulation of the osteoclast-supporting activity by TGF-β1 is mediated by activation of smad2 and smad3 signaling induced by interaction with the TGF-β receptors expressed on ST2 cells. In contrast to our findings, a previous study has reported that overexpression of *smad2* in the epithelium induced alveolar bone loss and increased the numbers of osteoclasts by upregulating TNF-α and RANKL [31]. It is possible that these inconsistent results may reflect differences in the cell type or stimulants. Among the growth factors of TGF-β family members, TGF-β1 and activin activate the intracellular signaling via smad2 and smad3 phosphorylation, while BMP group stimulates smad1, smad5, and smad8 [32–34]. Unexpectedly, our preliminary study showed

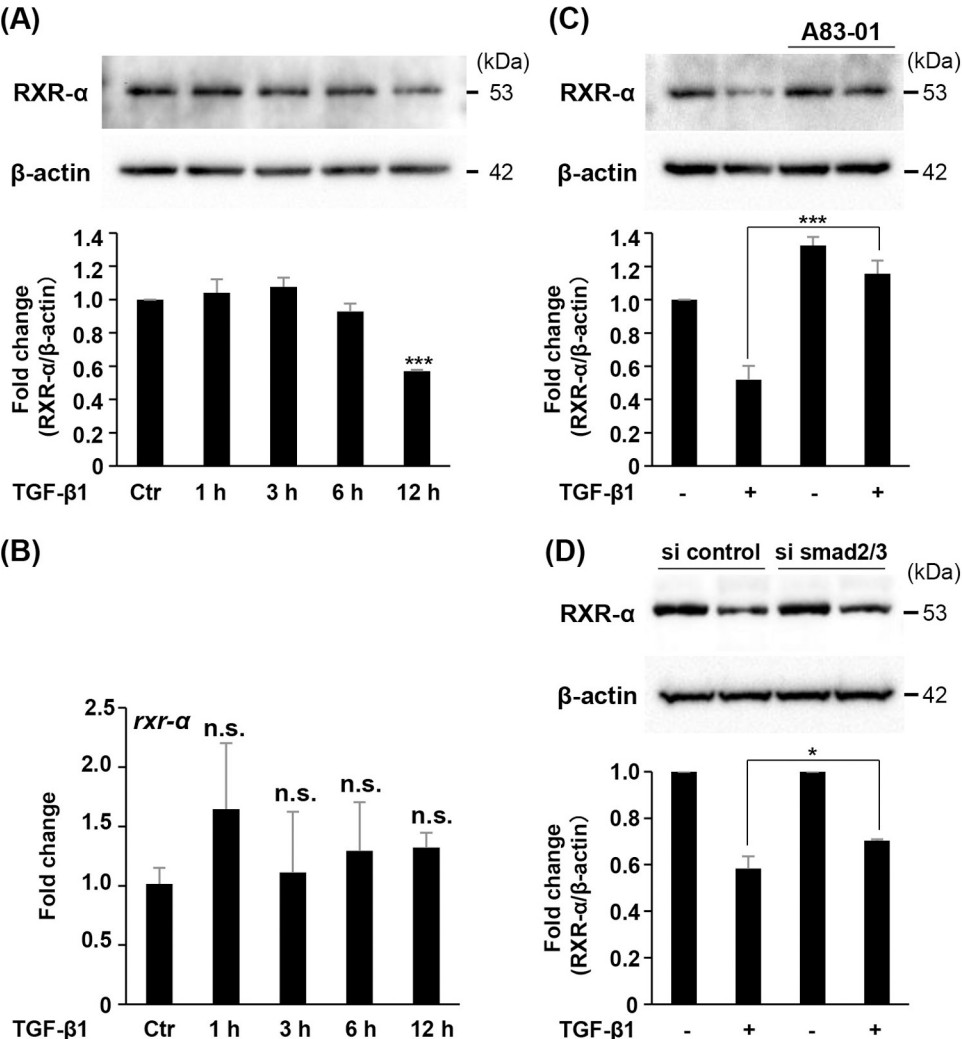

**Fig 5. Effect of TGF-β1 on RXR-α expression in ST2 cells.** (A, B) ST2 cells were treated with TGF-β1 (2.0 ng/mL) for indicated time periods. (A) RXR-α protein level was detected via Western blot analysis. β-actin served as the loading control. (B) The mRNA level of *rxr-α* was assessed via real-time RT-PCR. (C, D) ST2 cells were pretreated with A83-01 (2.0 μM) for 1 h (C) or transfected with either control siRNA or smad2/3 siRNA (D), prior to stimulation with TGF-β1 (2.0 ng/mL) for 12 h. RXR-α protein level was detected by Western blot analysis. β-actin served as the loading control. (A, C, D) Bars represent means with the standard deviation of relative band intensities normalized to changes in the β-actin from independent triplicate samples. Data are displayed as mean ± S.D. of three independent cultures. *n.s.* = not significant, $^*P < 0.05$, $^{***}P < 0.0001$.

that neither activin nor BMP-2 exerted an effect on RANKL expression induced by $D_3$ and Dex (data not shown). Further studies are needed to clarify the detailed regulatory mechanisms underlying the intracellular signal activation, such as the non-smad signaling pathway induced by TGF-β1 on RANKL expression.

The expression of the *Tnfsf11* is induced by $D_3$, parathyroid hormone (PTH), and interleukin-6-type (IL-6) via the activation of transcription factors, such as VDR, RXR, cAMP responsive element binding protein (CREB), and STAT3 [11]. The region of 76 kb upstream of the *Tnfsf11* transcriptional start sites (TSS) termed RL-D5 is essential to the regulation of RANKL induced by $D_3$, PTH, and IL-6 [18]. However, neither nuclear translocation of VDR nor phosphorylation of STAT3 protein, induced by $D_3$ and Dex, was affected by the TGF-β1 treatment.

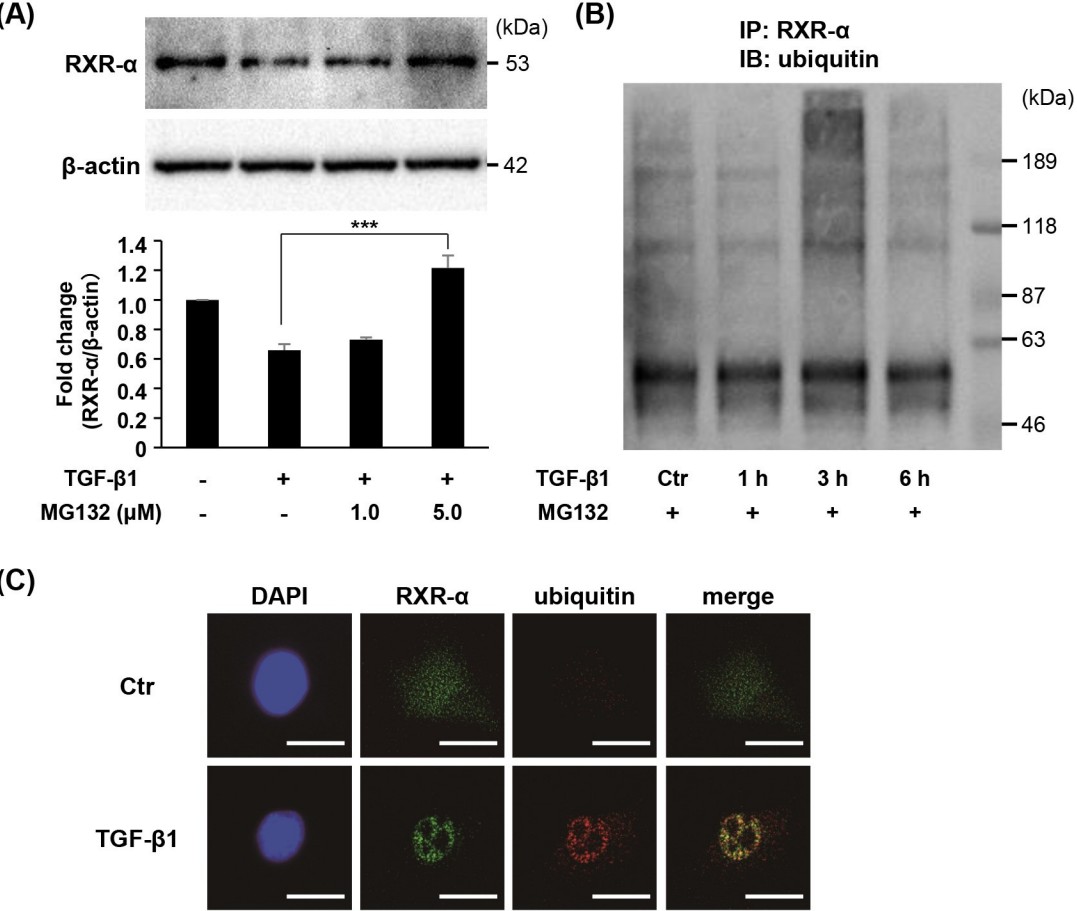

**Fig 6. Effect of TGF-β1 on RXR-α protein degradation mediated by ubiquitin-proteasome system.** (A) ST2 cells were pretreated with MG132 (1.0, 5.0 μM) for 1 h, followed by stimulation with TGF-β1 (2.0 ng/mL) for 12 h. The protein level of RXR-α was detected via Western blot analysis. β-actin served as the loading control. Bars represent means with the standard deviation of relative band intensities normalized to changes in the β-actin from independent triplicate samples. *** = $P < 0.0001$. (B) ST2 cells were pretreated with MG132 (5.0 μM) for 1 h prior to stimulation with TGF-β1 (2.0 ng/mL) for indicated time periods. Whole cell lysates were immunoprecipitated using anti RXR-α-conjugated magnetic beads. The resulting bound fraction was analyzed by Western blot analysis and probed for ubiquitin. (C) ST2 cells were treated with TGF-β1 (2.0 ng/mL) for 3 h. The cells were fixed and immunostained using monoclonal antibodies to detect RXR-α (green) and ubiquitin (red). Blue represents DAPI staining. Scale bars indicate 50 μm.

Thus, we focused on the RXR-α, which forms the complex with VDR and regulates the transcriptional activity. The expression of RXR-α protein but not mRNA was inhibited by TGF-β1. Furthermore, downregulation of RXR-α protein by TGF-β1 was recovered by A83-01 pretreatment and knockdown of smad2 and smad3. These data suggested that proteolytic event of RXR-α induced by TGF-β1-smad2/3 signaling is dependent on the suppression process of RANKL induced by $D_3$ and Dex. However, previous study reported that low concentration of TGF-β1 stimulates RXR expression via activation of activator protein 1 in osteoblastic MC3T3-E1 cells [35]. We have no ready explanation for these contrasting results, though it is possible that they reflect differences in the cell types or concentration of TGF-β1.

Recent studies have revealed that the ubiquitin-proteasome system is regulated by TGF-β1 and stimulates the regeneration of cancellous bone [36] and prostate cancer cell migration [37]. Thus, we focused on the ubiquitin-proteasome system as a candidate for the proteolytic signaling involved in protein degradation of RXR-α induced by TGF-β1. Proteasome inhibitor

treatment led to effective recovery of RXR-α protein levels which had been downregulated by TGF-β1. Moreover, poly-ubiquitination of RXR-α protein was induced by TGF-β1 treatment. These data suggested that TGF-β1 activated proteolytic signaling mediated by ubiquitin-proteasome system and led to the degradation of the RXR-α protein.

## Conclusion

This study demonstrated that TGF-β1 downregulated RANKL expression and osteoclast-supporting activity of osteoblasts/stromal cells, induced by $D_3$ and Dex, via degradation of RXR-α protein mediated by ubiquitin-proteasome system. TGF-β1 has a biphasic and complex effects on osteoclastogenesis. The effect of lower concentrations of TGF-β1 and the interaction of several intracellular signaling pathways activated by TGF-β1 in order to regulate RANKL expression in osteoblasts/stromal cells are currently under investigation in our laboratory. Furthermore, previous *in vivo* studies have been demonstrated that TGF-β1 acts as a coupling factor which communicate bone formation and resorption [38, 39]. To elucidate the temporal and spatial role of TGF-β1 in bone remodeling process, analyses using *in vivo* animal model are needed. Looking forward, extensive studies may help elucidate the biological effect of TGF-β1 in bone remodeling process.

## Supporting information

**S1 Raw images.**
(PDF)

## Acknowledgments

We thank Editage (www.editage.com) for editing a draft of this manuscript.

## Author Contributions

**Conceptualization:** Wataru Ariyoshi.

**Data curation:** Momoko Inoue, Yoshie Nagai-Yoshioka, Ryota Yamasaki, Tatsuo Kawamoto, Tatsuji Nishihara, Wataru Ariyoshi.

**Formal analysis:** Momoko Inoue, Yoshie Nagai-Yoshioka, Ryota Yamasaki, Tatsuo Kawamoto, Tatsuji Nishihara, Wataru Ariyoshi.

**Investigation:** Momoko Inoue, Yoshie Nagai-Yoshioka, Ryota Yamasaki, Tatsuo Kawamoto, Tatsuji Nishihara, Wataru Ariyoshi.

**Supervision:** Tatsuji Nishihara, Wataru Ariyoshi.

**Writing – original draft:** Momoko Inoue, Wataru Ariyoshi.

**Writing – review & editing:** Momoko Inoue, Yoshie Nagai-Yoshioka, Ryota Yamasaki, Tatsuo Kawamoto, Tatsuji Nishihara, Wataru Ariyoshi.

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
