## [Decision Letter · Decision Letter 0]

2 Nov 2021

PONE-D-21-26829Mechanisms involved in suppression of osteoclast supportive activity by transforming growth factor-β1 via the ubiquitin-proteasome systemPLOS ONE

Dear Dr. Ariyoshi,

Thank you for submitting your manuscript to PLOS ONE. After careful consideration, we feel that it has merit but does not fully meet PLOS ONE’s publication criteria as it currently stands. Therefore, we invite you to submit a revised version of the manuscript that addresses the points raised during the review process.Please discuss the in vitro and in vivo effects of TGF-beta1 on osteoclasts and emphasize what new information this study contributes to the field.

We look forward to receiving your revised manuscript.

Kind regards,

Xing-Ming Shi, Ph.D

Academic Editor

PLOS ONE

Journal Requirements:

3. Please ensure that you refer to Figure 2 in your text as, if accepted, production will need this reference to link the reader to the figure.

Reviewers' comments:

Reviewer's Responses to Questions

**Comments to the Author**

1. Is the manuscript technically sound, and do the data support the conclusions?

Reviewer #1: No

Reviewer #2: Yes

2. Has the statistical analysis been performed appropriately and rigorously? 

Reviewer #1: Yes

Reviewer #2: Yes

3. Have the authors made all data underlying the findings in their manuscript fully available?

Reviewer #1: Yes

Reviewer #2: Yes

4. Is the manuscript presented in an intelligible fashion and written in standard English?

Reviewer #1: No

Reviewer #2: Yes

5. Review Comments to the Author

Reviewer #1: Comment 1) Effects of TGF-beta1 in vivo and in vitro.

In this study, the authors demonstrated TGF-beta 1 down-regulated RANKL expression in vitro.

TGF-beta1 is well known to exert suppressive effects in vitro, and to exert promotive effects in vivo.

Actually, the many manuscript has been published already regarding TGF-beta 1 derived from bone matrix promoted osteoclastgenesis.

Therefore, this study demonstrated one-sided effects of TGF-beta 1 and effects of TGF-beta 1 is already known to the potential readers.

The authros should revise the manuscript extensively so that the potential readers can find new findings, and recognize this study does not contain one-sided effects.

Comment 2) Stromal cells or mesenchymal stem cells ?

Comment 3) Many sentences involving unclear words and meanings

Reviewer #2: This manuscript described the important role of the transforming growth factor-β1 (TGF-β1) on the osteoclast differentiation. In this paper, TGF-β1 downregulates RANKL expression and the osteoclast-supporting activity of osteoblasts/stromal cells through the degradation of the RXR-α protein mediated by ubiquitin proteasome system.

It should be discussed in detail whether the findings clarified in this paper can explain the mechanism of action of TGF-β1 on osteoclast differentiation by bone resorption factors other than vitamin D, PTH and PGE2.

6. PLOS authors have the option to publish the peer review history of their article (what does this mean?). If published, this will include your full peer review and any attached files.

Reviewer #1: No

Reviewer #2: **Yes: **Nobuyuki Udagawa

---

## [Author Response · Author response to Decision Letter 0]

16 Nov 2021

Comment 1) Effects of TGF-β1 in vivo and in vitro.

In this study, the authors demonstrated TGF-β1 down-regulated RANKL expression in vitro. TGF-β1 is well known to exert suppressive effects in vitro, and to exert promotive effects in vivo. Actually, the many manuscripts have been published already regarding TGF-β1 derived from bone matrix promoted osteoclastgenesis. Therefore, this study demonstrated one-sided effects of TGF-beta 1 and effects of TGF-β1 is already known to the potential readers. The authors should revise the manuscript extensively so that the potential readers can find new findings, and recognize this study does not contain one-sided effects.

We appreciate your important suggestions. Several studies demonstrated that the role of TGF-β1 on bone remodeling process. However, the reported effects of TGF-β1 are controversial. Since many of these studies are focused on the direct effects of TGF-β1 on osteoclast progenitors, we investigated the effects of TGF-β1 on RANKL generation in stromal cells. Moreover, the role of TGF-β1 as a coupling factor of bone remodeling process should be elucidated. However, we did not mention about it. We edited the Discussion session in the revised manuscript with some references.

Comment 2) Stromal cells or mesenchymal stem cells? 

In general, stromal cells involves all the non-epithelial cells including fibroblast, lipocytes, lymphcytes, muscular cells, and neuron cells. Only in the field of regeneration, a stromal cell is referred to as a mesenchymal stem cell (MSC). To avoid misunderstanding of the potential readers, the stromal cells in the current study should be revised as mesenchymal stem cells.

We appreciate your helpful suggestions. ST2 cell is a cloned stromal cell line from mouse bone marrow. This cell line has the ability to differentiate into osteoblast-like cells and shows high expression of RANKL and low expression of OPG when treated with 1,25(OH)2D3 and dexamethasone. However, we did not mention about it. So, we added this information in the Materials and methods section of revised manuscript.

Comment 3) Many sentences involving unclear words and meanings. 

There are many sentences involving unclear words and meanings. These sentences should be revised before submission. The following are the examples.

We appreciate your helpful comments. We carefully checked and revised the sentences that were pointed out.

Example 1) The current study investigated the effect of TGF-β1 on receptor activator of nuclear factor kappa-B ligand (RANKL) expression, induced by 1α,25(OH)2D3 (D3) and dexamethasone (Dex) in stromal cells. This is very complex. RANKL expression in stromal cell

Thank you for your helpful comments. We edited the abstract section of revised manuscript. 

Example 2) Stromal cells support osteoclastogenesis.; Please explain the meaning of "support osteoclastogenesis” in this study. Is this mean "induce osteoclastogenesis"? What is the difference? 

Thank you for your suggestions. Previous studies have been revealed the direct effect of TGF-β1 on the osteoclast differentiation from osteoclast progenitor cells. In this study, we showed that TGF-β1 indirectly inhibits the expression of RANKL, an accelerator of osteoclast differentiation in stromal cells. To emphasize these differences, we used the term “support osteoclastogeneis” in the place of “induce osteoclastogenesis”.

Example 3) ST2 cells in the abstract section. Please explain what ST2 cells mean in the abstract section.

As noted by the reviewer’s comment, we added the information of ST2 cells in the abstract section. 

Example 4) Transforming growth factor-β1 (TGF-β1) is a coupling factor vital for bone remodeling. What means "vital bone remodeling"?

As noted by the reviewer’s comment, we edited the abstract section of revised manuscript.

---

## [Editor Report · Decision Letter 1]

30 Dec 2021

Mechanisms involved in suppression of osteoclast supportive activity by transforming growth factor-β1 via the ubiquitin-proteasome system

PONE-D-21-26829R1

Dear Dr. Ariyoshi,

We’re pleased to inform you that your manuscript has been judged scientifically suitable for publication and will be formally accepted for publication once it meets all outstanding technical requirements.

Kind regards,

Xing-Ming Shi, Ph.D

Academic Editor

PLOS ONE
---

## [Editor Report · Acceptance letter]

14 Feb 2022

PONE-D-21-26829R1 

Mechanisms involved in suppression of osteoclast supportive activity by transforming growth factor-β1 via the ubiquitin-proteasome system

Dear Dr. Ariyoshi:

I'm pleased to inform you that your manuscript has been deemed suitable for publication in PLOS ONE. Congratulations! Your manuscript is now with our production department. 

Kind regards, 

on behalf of

Dr Xing-Ming Shi 

Academic Editor

PLOS ONE